# Measurement Methods of Fatigue, Sleepiness, and Sleep Behaviour Aboard Ships: A Systematic Review

**DOI:** 10.3390/ijerph19010120

**Published:** 2021-12-23

**Authors:** Fiona Kerkamm, Dorothee Dengler, Matthias Eichler, Danuta Materzok-Köppen, Lukas Belz, Felix Alexander Neumann, Birgit Christiane Zyriax, Volker Harth, Marcus Oldenburg

**Affiliations:** 1Institute for Occupational and Maritime Medicine Hamburg (ZfAM), University Medical Center Hamburg-Eppendorf (UKE), 20459 Hamburg, Germany; fiona.kerkamm@web.de (F.K.); dorothee.dengler@justiz.hamburg.de (D.D.); mail@dreichler.de (M.E.); danuta.materzok@atem-balance.de (D.M.-K.); lukas.belz@justiz.hamburg.de (L.B.); volker.harth@justiz.hamburg.de (V.H.); 2Midwifery Science-Health Services Research and Prevention, Institute for Health Services Research in Dermatology and Nursing (IVDP), University Medical Center Hamburg-Eppendorf (UKE), 20246 Hamburg, Germany; Fe.neumann@uke.de (F.A.N.); b.zyriax@uke.de (B.C.Z.)

**Keywords:** actigraphy, fatigue, measurement method, polysomnography, pupillometry, seafaring, sleep, sleepiness

## Abstract

Since seafarers are known to be exposed to numerous job-related stress factors that can cause fatigue, sleepiness, and disturbed sleep behaviour, the aim of this review was to provide an overview of the subjective and objective measurement methods of these strains. Using a systematic review, 166 studies were identified within the period of January 2010 to December 2020 using the PubMed database. Of the 21 studies selected, 13 used both subjective and objective measurement methods. Six studies used only subjective and two studies only objective methods. For subjective assessment, 12 different questionnaires could be identified as well as activity and sleeping logs. Actigraphy and reaction time tests (RTT) were the most common objective methods. In single cases, electrooculography (EOG), pupillometry and ambulatory polysomnography (PSG) were used. Measurement-related limitations due to vessel-related impacts were less often reported than expected. No restrictions of daily routines on board were described, and only single-measurement disturbances due to ship movements were mentioned. The present literature review reveals that there are various routines to measure fatigue, sleepiness, and sleep behaviour on board. A combination of subjective and objective methods often appears to be beneficial. The frequent use of actigraphy and RTT on board suggests good feasibility and reliable measurements with these methods. The use of ambulatory PSG in maritime-like contexts suggests that this method would also be feasible on board.

## 1. Introduction

Seafarers are often exposed to psychophysical stress due to isolation, separation from family, time pressure, and long working days. In addition, noise from ship operations, vibration from the engine, and weather-related ship motion are other significant stressors that can reduce sleep quality on board [1]. Moreover, 91.6% of seafarers stated that they are frequently disturbed by at least one environmental factor, such as excessively cold or warm ambient temperatures, odours, noise, poor bedding conditions, or ambient light when sleeping in their cabins [2]. Noise was the most frequently mentioned disturbing stressor in relation to sleep, with 62.4% of seafarers complaining, followed by ambient temperature (57.3%).

Overall, the causes of fatigue, sleepiness, and disturbed sleep on board are varied. The International Maritime Organization (IMO) [3] defines fatigue as a psychophysical consequence of a stressful situation typical of shipping that can negatively affect ship safety. The effects of fatigue are said to be manifested, for example, in reduced attention and memory, lowered responsiveness, increased risk-taking, and reduced problem-solving ability [3]. Fatigue and sleepiness can thus increase the risk of accidents at sea. In line with that, an analysis of 44 Incident at Sea Reports (with human errors as contributing factor) found that 86% of the analysed accidents had references to sleep—most of these (34%) were connected to “sleep loss as a way of life” (waking at odd hours, daytime sleep, working instead of sleeping, and sleep hygiene factors) [4]. A review of additional 279 maritime accidents showed that fatigue contributed to 16% of critical ship incidents and 33% of personnel injuries [5].

Some of the ship’s personnel (particularly the nautical officers and watchkeeping crew deck ranks) work 24-h shifts. These alternating day/night shifts cause disturbances of the circadian rhythm. Shift systems, such as the 6:6 and 4:8 shift system, only allow short interruptions in work for recovery and sleep phases. In the case of the 6:6 system, a watchkeeper works for six hours and then has six hours of free time—in constant rotation with the second watchkeeper. Shift changes usually take place at 6:00 a.m., 12:00 p.m., 6:00 p.m., and midnight. During the 4:8 system, three nautical officers take turns, each working four hours, followed by an eight-hour rest period. Shift changes occur at 4:00 a.m., 8:00 a.m., 12:00 p.m., 4:00 p.m., 8:00 p.m., and midnight. Especially during night work of the 6:6 system (midnight to 6 a.m.), increased sleepiness and shorter sleep episodes could be found on board ships [6].

While a variety of subjective and objective methods for measuring sleepiness exists, the determination of fatigue has so far been based only on subjective methods [7]. Even though sleepiness—unlike fatigue—can therefore be measured, e.g., using the Multiple Sleep Latency Test (MSLT), a clear definition and distinction between the terms fatigue and sleepiness is difficult, and therefore, these terms are often used interchangeably in publications [8]. Currently, there is a debate as to whether sleepiness can be considered a sub-component of fatigue [9] or whether the terms refer to two different constructs [10]. In the literature, sleepiness is often described as reduced central nervous activation, which can be attributed primarily to little or non-restorative sleep and has no correlation to psychological causes [11]. Fatigue, on the other hand, is described as a general feeling of subjective exhaustion and reduced performance, which can be caused by physical, psychological, and cognitive stress. However, unlike sleepiness, daytime sleep episodes are atypical in fatigue [11].

Overall, an objectifying survey of fatigue, sleepiness, and seafarers’ sleep behaviour is an important component for seafarers’ health and for the safety on board. Especially considering that fact, Allen et al. [12] concluded in their review “Seafarers’ fatigue: a review of the recent literature” that the prevalence of fatigue in the maritime context appears to be higher than the seafaring industry is capable of or prepared to measure. Since a certain number of studies have already examined these strains using a wide variety of methods, this review aims to provide an overview of the objective and subjective measurement methods of fatigue, sleepiness, and sleep behaviour used on board. Furthermore, this work intends to provide support in the selection of measurement methods for future maritime studies.

## 2. Materials and Methods

As part of the interdisciplinary project “e-healthy ship”—which aims to optimise health management on board without the presence of doctors—a systematic literature search of maritime field studies covering the observation period from January 2010 to December 2020 was conducted using the PubMed database. Studies were independently screened for eligibility by three reviewers. Studies on fatigue or sleep in seafarers were identified using the following search terms or MeSHTerms (Medical Subject Headings): (sailor*[Title/Abstract] OR seafarer*[Title/Abstract] OR seamen[Title/Abstract] OR seaman [Title/Abstract] OR naval[Title/Abstract] OR ship[Title/Abstract] OR shipping[Title/Abstract] OR ships[Title/Abstract] OR maritime[Title/Abstract]) AND (OSA[Title/Abstract] OR apnoea*[Title/Abstract] OR sleep*[Title/Abstract] OR PSG[Title/Abstract] OR polysomno*[Title/Abstract] OR fatigue*[Title/Abstract]).

The search terms only included “seaman/-men” and not “seawoman/-women” due to the fact that female forms did not generate additional hits, as seafaring continues to be a male-dominated occupation.

This search string yielded a total of 166 hits during the above-mentioned observation period. Appropriate studies were selected according to the PRISMA (Preferred Reporting Items for Systematic Reviews and Meta-Analyses) statement (Figure 1) [13].

The following inclusion criteria were established: field studies in English or German; study population ≥ 10; seafarers were primarily assessed for fatigue, sleepiness, or sleep behaviour while sleeping on board; and use of standardized measurement methods.

After screening the abstracts, 145 studies were excluded since they did not meet the inclusion criteria (*n* = 97) or were not thematically related to seafaring (*n* = 48).

After full-text review, two additional studies were excluded because the seafarers predominantly did not sleep on board [14,15]. In addition, two studies were hand-selected (one of these studies included tests in a ship simulator [16], and the other had a participant number of only eight sailors [17]). Although these papers were initially eliminated according to exclusion criteria, they were included due to the use of polysomnography (PSG) as a medical standard for sleep assessment in a maritime context.

Finally, a total of 21 studies on fatigue, sleepiness, and sleep behaviour on board were included in the review. From the identified studies, information on the aim of the study; the study population; the methods of measurement of fatigue, sleepiness, and sleep behaviour of seafarers on board; and limitations of these measurement methods were summarised. In addition, the evidence levels of these studies were evaluated by the Scottish Intercollegiate Guidelines Network criteria [18].

## 3. Results

The selected 21 studies were maritime field studies, apart from the manually selected study by van Leeuwen et al. [16], which used a bridge simulator. The studies were conducted in the period from 2002 to 2018, with only ten studies reporting a specific study year. Overall, the evidence level of these studies ranged from 2− to 2+, with eight studies meeting the SIGN criterion of 2− and 13 studies corresponding to the criterion of 2+.

Populations varied from eight to 1269 subjects (median 84 subjects). Analysis of the studies’ methods of measuring fatigue, sleepiness, and sleep behaviour revealed that 13 studies used both subjective and objective measurement methods. Six studies used only subjective tests, and two studies used only objective methods (Table 1).

### 3.1. Subjective Measurement Methods

Twelve different questionnaires (overview in Table 2), each with a different focus, as well as activity and sleep diaries were used to subjectively assess fatigue, sleepiness, and sleep behaviour (Table 1).

Fatigue was assessed by the Fatigue Severity Scale (FSS; *n* = 1) and the Swedish Occupational Fatigue Inventory (SOFI; *n* = 3) as well as the Samn–Perelli Fatigue Scale (SPFS; *n* = 1), which is also known as the Crew Status Survey (CSS). Fatigue was also assessed in the Profile of Mood States (POMS; *n* = 3) in conjunction with six other mood states. Work-associated fatigue was evaluated using the Need for Recovery Scale (NFR; *n* = 1).

Sleepiness was assessed using the Epworth Sleepiness Scale (ESS; *n* = 6), the Karolinska Sleepiness Scale (KSS; *n* = 2), and the Stanford Sleepiness Scale (SSS; *n* = 1).

Sleep quality was assessed using the Pittsburgh Sleep Quality Index (PSQI; *n* = 4). Insomnia was assessed using the Insomnia Severity Index (ISI; *n* = 2) and the Bergen Insomnia Scale (BIS; *n* = 1). Possible sleep disorders were excluded using the Karolinska Sleep Questionnaire (KSQ; *n* = 1).

Exact differentiation of the meaning of activity and sleep diaries appeared to be difficult, as the terms were often used synonymously. A comprehensive explanation of which data were collected at which intervals was often not provided.

### 3.2. Objective Measurement Methods

A total of seven different objective measurement methods were recorded in the selected studies. Of these, actigraphy was by far the most commonly used to determine sleep duration and quality. A total of nine different devices from five different manufacturers were used in 13 studies although only the manufacturer (not device designation) was reported in the study by Thomas et al. [32]. In this study selection, a tendency towards devices from Philips Respironics (Actiwatch Spectrum = 2; Actiwatch Spectrum Plus = 2; Actiwatch = 1; Actiwatch 2 = 1; no model specified = 1) and the Motionlogger Watch from Ambulatory Monitoring (*n* = 6) emerged. In addition, devices from ActiGraph (GT9X Link = 1; GT3X = 1), Cambridge Neurotechnology (Actiwatch = 1; Actiwatch AW4 = 1), and BodyMedia (SenseWear armband activity monitor = 1) were used. A comparison of these actigraphy devices revealed that five of the ten actigraphs were currently still offered for sale by the respective manufacturers (as of December 2021) (see Appendix A, Table A1).

The second most used method was a reaction time test (RTT) to objectively measure drowsiness-related limitations (*n* = 7). Twelve studies used the reaction time test procedure of the psychomotor vigilance test (PVT).

Ambulatory polysomnography (PSG) was used in two studies. Léger et al. [17] used PSG on sailing yachts during a race to measure TST (total sleep time) and TSD (total sleep debt) on board. A night-cap (a head actigraph with EOG) was used beforehand to determine TST. In the study by van Leeuwen et al. [16], the PSG was used in a bridge simulator to record sleepiness during watch.

Furthermore, Lützhöft et al. [6] captured sleepiness using EOG measurements, and Oldenburg and Jensen [27] investigated the effects of sleepiness using pupillometry. Jaipurkar et al. [24] additionally used pulse and blood pressure measurements to compare the persons’ physical changes on land and at sea.

A few studies determined parameters about the onboard sleep environment. First, noise dosimeters [31] and wet bulb thermometers [24] were used. Second, environmental conditions (e.g., temperature, noise, light, air quality, odours, ventilation, and ship motion) were subjectively surveyed [2].

## 4. Discussion

The review included 21 studies that examined fatigue, sleepiness, and sleep behaviour of seafarers in their shipboard work environment. It was found that many different subjective and objective measurement methods have already been used on board for this purpose. Although few acute measurement problems were reported in the selected studies, general limitations of these methods should also be considered. A summarising overview of the generally known strengths and weaknesses of each method is given in Table 3.

### 4.1. Subjective Measurement Methods and Limitations

#### 4.1.1. Questionnaires

Questionnaires were used as subjective testing methods in 15 of the 21 studies. Besides lack of motivation, compliance [6], and recall bias, the factor of social desirability plays a limiting role [16]—especially when respondents fear negative reactions from their superiors when indicating fatigue or exhaustion. This could be relevant, for example, in the context of temporary contracts, which are commonly used in the shipping industry [22].

Moreover, the presence of interviewers could have a negative impact on responses. In a maritime study by Bridger et al. [19], there was a significant increase in Need for Recovery (NFR) scores during the second questionnaire survey, which was conducted without an interviewer, unlike the initial survey. However, Bridger et al. [19] suggested that the increased scores could also be due to the fact that seafarers affected by fatigue were more likely to participate in the second survey as well.

Furthermore, there is the possibility that the multicultural seafarer sample has a crucial impact on the response behaviour. On the one hand, misunderstanding may occur due to difficulties in understanding because of language barriers or low educational level. On the other hand, intercultural differences may influence the individual interpretation of the questions [33].

In general, questionnaires offer a variety of ways to subjectively measure fatigue, sleepiness, or sleep problems. Existing reviews offer an overview to find the appropriate test for the respective research question [7,35]. Particular attention should be paid to the distinction between fatigue and sleepiness. This was clarified by Matsangas and Shattuck [10] comparing the correlation of the ESS and FSS questionnaires of U.S. Navy crew members. They found that subjects’ subjective fatigue did not necessarily correlate with subjective sleepiness, so these should be considered as different constructs.

Moreover, it should be taken into account that questionnaires were generally originally designed for the land context and not for the use at sea. For example, the ESS questionnaire includes a question on road traffic, which cannot be answered by all seafarers from personal experience [36].

In addition, different questionnaire cut-off values might complicate a uniform evaluation. For example, Matsangas and Shattuck [10] reported different FSS cut-offs ranging from >3 to ≥5.4 (Table 3).

#### 4.1.2. Diaries

In 10 of the 21 studies, sleep or activity diaries were used. An advantage of this method is the flexible design, which can be individually adapted to the study design. Nevertheless, this fact makes it difficult to compare the studies with each other. According to Quante et al. [37], there is no standardized format for an activity diary. However, recommendations for sleep diaries exist in English-speaking [38] as well as in German-speaking countries [11].

Using diaries alone to determine sleep duration and quality is not recommended as their results are often subjectively overestimated [20,24]. However, in two of the selected studies, diaries are reported as a helpful adjunct to actigraphy to supplement missing actigraphy data [25,30]. Supplementing sleep measurements with diaries could be particularly useful for shipboard measurements if actigraphy data cannot be analysed, e.g., due to heavy ship-movements.

### 4.2. Objective Measurement Methods, Differences and Limitations

In the included 21 studies, objective measurement methods were applied in 15 cases. None of these studies mentioned that the use of an objective measurement method would have significantly disrupted the daily routines, and thus, the routine activities of the seafarers on board and only a few limitations were attributed to maritime causes.

#### 4.2.1. Actigraphy

Actigraphy was the most commonly used objective measurement method on board in the identified studies. It represents a validated method for assessing sleep and wake patterns in field studies over long periods of time [25]. As a limitation, it can be seen that measurements that require specialised, complex equipment are less suitable for simultaneous investigation of large study populations (Table 3). For example, Jaipurkar et al. [24] stated that not all 50 subjects on board could be equipped with actigraphs at the same time. In addition, objective measurements require the presence of investigators on board. This results in a more laborious and expensive implementation and may also cause a Hawthorn effect [39]. Furthermore, actigraphic measurements can be susceptible to ship movements. Sunde et al. [31] had to exclude 6% of the actigraphy recordings because a sea state >3 as well as very high speeds affected the reliability of the actigraph on board.

#### 4.2.2. Electrooculography (EOG)

It should additionally be noted that even objective measurement methods are not entirely free of interindividual differences. Although an association between sleepiness and increased blink measurements in the EOG is assumed, Lützhöft et al. [6] indicated that blink characteristics vary for people individually. Therefore, EOG measurements should not be used exclusively to determine sleepiness.

Additionally, the measurement environment may negatively affect the results. The measurement problems of objective methods reported in the present study selection suggest that increased vessel motion may influence actigraphy and EOG results [6,31]. However, an underreporting of similar measurement inaccuracies in other studies can be presumed. It is assumed that actigraphy, EOG, and correspondingly PSG measurements can be well performed when extreme weather conditions do not occur.

#### 4.2.3. Pupillometry

Conducting pupillometry measurements with completely darkening goggles, as in the study by Oldenburg and Jensen [27], is necessary because ambient light conditions have been shown to affect pupil parameters during pupillometry [40]. In addition, strong vessel motion or noise disturbances could impact pupillometric measurements, as we can report from our own experience that subjects can be distracted, and measurements can be biased as a result.

#### 4.2.4. Reaction Time Tests (RTT)

Such distracting environmental conditions could also have a negative effect on the reliability of an RTT measurement, as this test requires the undivided attention of the subjects. In addition, it is recommended that the time of day and the awake time of the subjects should be taken into account when performing an RTT. Reaction time may be significantly slowed when testing does not coincide with natural biological sleep-wake cycles, as is often the case with shift workers (e.g., on board) [41].

#### 4.2.5. Polysomnography (PSG)

In contrast to actigraphy, the use of ambulatory PSG aboard ships has not been adequately studied to date, according to our research. However, studies from comparable workplace-specific contexts showed that reliable polysomnographic measurements in unusual environments are indeed possible.

Measurements were conducted either in a bridge simulator [16] or during a yacht race [17]. Jay et al. [42] studied the sleep of train drivers who worked an 8:8 shift system and slept in so-called “relay vans” on board. The authors emphasized the benefits of PSG over previous actigraphy studies, as it allowed them to evaluate new aspects of sleep quality. Because of the shift work associated with movement, vibration, and noise of the sleep environment, this study design shows parallels to maritime field studies. Polysomnographic measurements in an environment characterized by noise and turbulence have likewise been used on airplanes [43]. In the study by Mitler et al. [44], truck drivers were monitored while driving using EOG and camera recordings. In addition, their sleep was recorded using PSG in accommodations along their route. Mairesse et al. [45] performed wireless PSG measurements with two-channel EEG devices at the Antarctic research station “Concordia”, which yielded good results despite limited PSG set-up.

In these four studies, no significant interference of ambulatory PSG by the unusual measurement environments was mentioned. This fact suggests that reliable polysomnographic measurements in unusual environments—which may well resemble conditions on board ships—can be possible without major interference.

### 4.3. Comparison of Actigraphy and PSG

In contrast to PSG, the use of actigraphs is considered cost effective and non-disruptive (Table 3). Actigraphic measurements are less disturbing to sleep architecture than PSG and are unlikely to be limited by a first night effect [46]. In addition, actigraphy is an established method in field studies investigating circadian rhythms and is also indicated in the assessment of individuals with shift work disorders [47].

According to Quante et al. [37], there is no universal minimum duration of an actigraphy measurement to obtain the most reliable results. The Standards of Practice Committee of the American Academy of Sleep Medicine recommends a minimum of three consecutive 24-h actigraphy measurements to obtain reliable sleep-wake estimates [48]. Nevertheless, in the International Classification of Sleep Disorders (ICSD-3), a minimum duration of actigraphy measurements over seven days in combination with diaries is usually recommended. In case of circadian rhythm disturbances—which are of particular interest in the maritime context—actigraphy should be conducted for at least 14 days [49].

However, actigraphy is not suitable for studying sleep architecture because it is based only on accelerometery [50]. Thus, sleep can easily be overestimated and wakefulness underestimated [51]. This low specificity for detecting waking phases during sleep is reflected in lower validity in individuals with poor sleep quality. In addition, awake phases also increase with age so that the accuracy of actigraphy decreases age dependently [52]. Despite these limitations, studies with young and healthy subjects showed 91–93% agreement between actigraphy and PSG in common measurement parameters, such as total sleep time or sleep efficiency [11]. However, it should be kept in mind that the correlation of actigraphy and PSG parameters is significantly dependent on the placement of the actigraph [46]. Thus, placement of the actigraph at the wrists showed better agreement than measurements via hip sensors.

Overall, stationary PSG represents the gold standard of sleep diagnostics. In contrast to actigraphy, it is suitable for the diagnosis of sleep-related breathing disorders, such as obstructive sleep apnoea syndrome [53]. It has been proven that ambulatory PSG achieved similar results to stationary PSG in comparative studies [54,55]. Whether a first-night effect also occurs in ambulatory PSG has not been conclusively established [55,56,57,58].

### 4.4. Comparison of Subjective and Objective Measurement Methods

In the literature, a rather low correlation of subjective and objective sleep or sleepiness measurement methods is assumed. While objective methods capture physiological aspects of sleepiness, subjective data depend on the introspection ability of the subjects [11]. The study by Oldenburg and Jensen [27] also showed only a weak correlation of subjective Stanford Sleepiness Scale (SSS) values with pupillometry data. It was striking that especially the younger, more inexperienced seafarers self-assessed their sleepiness more severely than this could be objectified by pupillometry. Here, the authors indicated that pupillometry had not yet been an established and universally accepted screening method for measuring sleepiness. In other studies, however, pupillometry was meanwhile described as established and widely used [59,60].

Youn and Lee [34] based their choice of method on the fact that objective measurement methods are superior to subjective measurement methods in terms of feasibility, validity, and reliability. Matsangas and Shattuck [25] found an association of Pittsburgh Sleep Quality Index (PSQI) scores > 9 with elevated PVT scores. They suggested screening risk groups using questionnaires and examining only subjects at increased risk more closely by using objective methods.

In general, we found that subjective and objective methods were often used complementarily. Therefore, it seems reasonable to choose a combined study design and, for example, complement actigraphy measurements with sleep diaries [61]. Further studies should be conducted to directly compare subjective and objective measurement methods for fatigue, sleepiness, or sleep behaviour in a maritime context.

### 4.5. Influences of Stress on Fatigue, Sleepiness, and Sleep Behaviour

Two of the studies focused on the relationship of stress to fatigue, sleepiness or sleep behaviour.

Hystad and Eid [22] were able to show a correlation between disruptive environmental influences with reduced sleep quality and fatigue. Furthermore, low psychological capital (PsyCap) also predicted fatigue and poor sleep quality. PsyCap is a construct based on ideas from positive psychology defined as an individual’s level of high self-efficacy, optimism, hope, and resiliency. The findings on PsyCap are particularly interesting as PsyCap is seen as a changeable and developmentally characteristic. Interventions to develop PsyCap could therefore be a relatively simple and cost-effective way for maritime organisations to address fatigue and poor sleep quality in their employees.

Moreover, Thomas et al. [32] investigated the effects of increased workload of roll-on roll-off ferry workers during the summer “double sailing period”. Since participants had the opportunity to adjust their sleep duration to the increased workload, subjective assessment of fatigue as well as neurological performance were not negatively affected. Accordingly, this study outlined the adaption of sleep duration to changing working hours as an example of a functioning fatigue-risk management system, which can help maintain sleep and performance during periods of increased work intensity.

### 4.6. Further Factors That Should Be Considered in Measurements of Fatigue, Sleep, and Sleep Behaviour on Board

In addition to standardised questionnaires, individual parameters of seafarers should be taken into account. A survey of medication intake (especially of stimulating or sedative substances) as well as the seafarers’ general lifestyle (caffeinated beverages, alcohol consumption, physical fitness) seems advisable. For example, in their review on the use of addictive substances among seafarers, Pougnet et al. [62] found an increased prevalence of tobacco and alcohol use compared to the general population.

Additionally, it is useful to consider the individual circadian rhythm of seafarers. To determine the chronotype and thus the daily peak of attention, the Morningness-Eveningness Questionnaire (MEQ) is useful [63]. This questionnaire was applied in two of the selected studies [27,29].

Other important factors for sleep assessment relate to environmental conditions. Matsangas and Shattuck [2] reported that 91.6% of seafarers had their sleep disturbed by at least one environmental factor (noise, temperature, light, ship motion, odours, poor bedding). Indeed, another study demonstrated some subjective habituation to ship noise during sleep. However, actigraphic measurements did not confirm this subjective habituation effect [64]. During the study by Sunde et al. [31], noise was also associated with increased movements of seafarers during sleep and lower sleep efficiency. Therefore, especially in the maritime setting, it seems reasonable to incorporate the unusual sleep environment into the study design by, for example, measuring vibration, temperature, or noise to gain a better understanding if these factors may have disrupted sleep or sleep measurements.

In addition, measurements of fatigue, sleepiness, and sleep patterns on board should consider the respective variability for different shift systems, occupational groups, voyage episodes, and ship types. The guideline “Health aspects and design of night and shift work” (“Gesundheitliche Aspekte und Gestaltung von Nacht- und Schichtarbeit”) of the German Society for Occupational and Environmental Medicine [65] states that irregular shift times in particular are often associated with sleep deficits and fractional sleep periods. This was confirmed in the study by Arendt et al. [66], which investigated the sleep and circadian rhythms of seafarers. Among other things, they compared the effects of a constant 4:8 system with a weekly rotating schedule. It was found that weekly changes in waking times resulted in poorer sleep efficiency as well as fragmented sleep. Furthermore, divergent results were evident during various voyage episodes: port stay, river, or sea passage. Since working hours were usually longer during port stays, the crew had more free time and sleeping opportunity during sea passages. While seafarers slept an average of only 6.9 h in port, they were able to record 7.6 h of sleep during a river passage and 8.9 h at sea [67].

## 5. Limitations

As a limitation, it must be mentioned that the literature search was limited exclusively to German- and English-language publications. Furthermore, the comparison of the benefits of the individual measurement methods is impeded by the fact that some studies differed considerably in their study design as well as in the use of the various measurement methods (e.g., number, duration of measurement). A partly imprecise distinction between sleepiness and fatigue made the research more difficult. A comparison of objective and subjective methods took place only sporadically. In addition, very few studies stated that they had questioned interfering variables (e.g., ship movements, vibrations) during the study or that they had objectively recorded them or even adjusted them as influencing variables. For this reason, our results are partly based on assumptions due to missing information from the authors.

## 6. Conclusions

The possibilities of measuring fatigue, sleepiness, as well as sleep behaviour on board are various. Since the distinction of fatigue and sleepiness is not uniform in the literature, the choice of the best suitable measurement method is additionally complicated for the investigator. In general, our research showed that a combination of subjective and objective measurement methods can be beneficial. A wide range of questionnaires exists to investigate specific issues regarding fatigue, sleepiness, or sleep behaviour. Fatigue can only be determined subjectively. In this context, the most frequently used questionnaires were the Swedish Occupational Fatigue Inventory (SOFI) and the Profile of Mood States (POMS). Subjective sleepiness was most commonly assessed with the Epworth Sleepiness Scale (ESS). For the effect of sleepiness, reaction time measurements (RTT, PVT) as well as pupillometry can be used. To determine sleep behaviour over longer periods, the use of actigraphs in combination with diaries for a minimum period of three to seven days is recommended. To facilitate the consistent use of diaries, digital logs with reminder function, e.g., in the form of apps, could be used in the future. If an accurate determination of sleep parameters, a representation of sleep architecture or a diagnosis of sleep disorders, such as obstructive sleep apnoea syndrome, is requested, the use of an ambulatory PSG, which achieves comparable results to stationary PSG, is recommended.

In general, fewer measurement-related limitations on board were reported than expected. No disruptions of daily board routines were reported, and only single measurement disturbances due to ship movements were mentioned. The frequent use of actigraphy and PVT on board therefore implies good feasibility and reliable measurements with these methods. The use of ambulatory PSG in maritime-like contexts hints that this method would as well be feasible on board. However, a higher rate of unreported cases, especially with objective measurement problems, must be assumed. Thus, it cannot be extrapolated with certainty what impact the maritime environment actually has on the reliability of the measurement methods. Further studies on board are required for this. Nonetheless, this review provides a useful overview and orientation for future maritime studies. As the working and living conditions on board are unique and cannot be compared to those on land, occupation-specific studies in these demanding workplaces on board are essential and of high value. This study highlights the need to adjust the selection of measurement methods for fatigue, sleepiness, and sleep behaviour of seafarers to the specific conditions on board.

## Figures and Tables

**Figure 1 ijerph-19-00120-f001:**
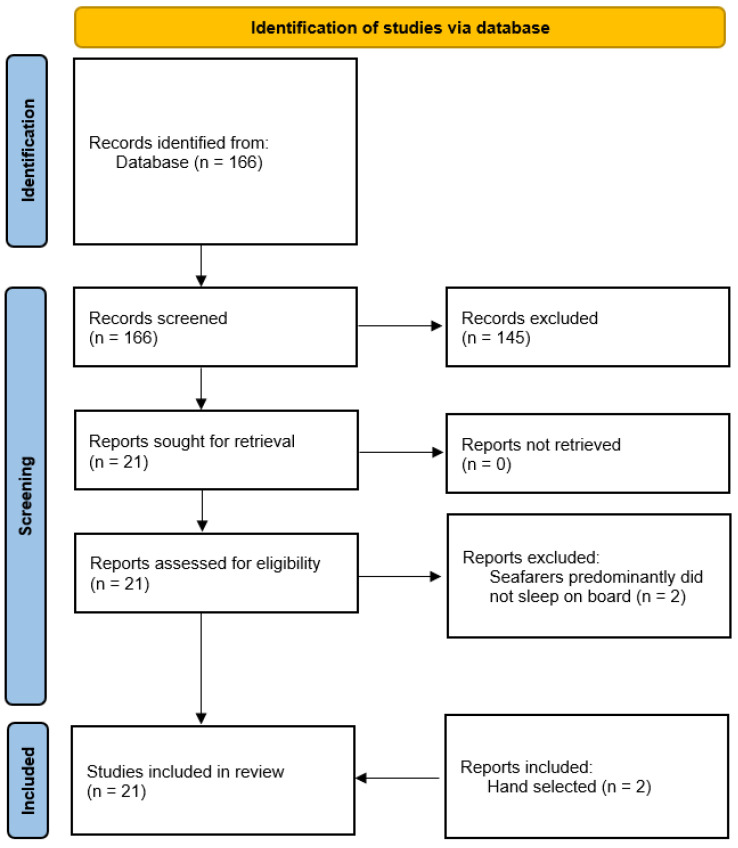
Search strategy on maritime studies of fatigue, sleepiness, and sleep behaviour following the PRISMA statement.

**Table 1 ijerph-19-00120-t001:** Characteristics and methods of the selected studies.

Author (Year)	SIGN-Criteria	Study Aim	Population	Year of Investigation	Measuring Method of Fatigue, Sleepiness, or Sleep Behaviour			
					Subjective	Objective		
						Actigraphy	RTT	Other
Bridger et al. (2010) [19]	2+	To investigate the relationship between age, job demands, and recreational needs in the maritime industry	322 employees on seven Royal Fleet Auxiliary vessels	November 2008–July 2009	NFR			
Harris et al. (2015) [20]	2−	To document characteristics of sleep disturbances; to examine the relationship between objectively derived and self-reported sleep indices and sleep quality	29 active-duty male Naval Special Forces		Sleep log	Motionlogger Watch (Ambulatory Monitoring, Inc., Ardsley, NY, USA)		
Hurdiel et al. (2014) [21]	2+	To evaluate sleep during solo offshore sailing races and compare reaction times during a reaction time test before and after these races	Twelve professional sailors on yachts (10-m Figaro 2 Beneteau)		Sleep log	GT3X (ActiGraph, Pensacola, FL, USA)	5-min PVT	
Hystad and Eid (2016) [22]	2+	To assess the effects of duration at sea, seafaring experience, environmental stressors, and psychological capital (PsyCap) on sleep quality and fatigue	742 seafarers (402 in the offshore supply industry and 340 aboard combined passenger roll-on/roll-off ferries and cargo ships) on 22 different vessels in the North Sea and Southeast Asia		SOFI (revised 20-item version) PSQI (only Ro-Ro-ferry)			
Hystad et al. (2013) [23]	2+	To investigate the effects of safety climate and psychosocial work environment on reported fatigue	402 seafarers working in offshore oil and gas industry on 22 vessels operating in the North Sea and Southeast Asia		SOFI (revised 20-item version)			
Jaipurkar et al. (2019) [24]	2−	To assess and compare sailors’ work-rest rhythms and alertness levels during sailing and non-sailing days; to compare ‘‘sleep duration’’ data as recorded in the sleep diary with actigraphy sleep data	32 male participants from a large Indian naval vessel		Sleep log Activity log	Actiwatch (Philips Respironics, Bend, OR, USA)	5-min PVT	Pulse und blood pressure
Léger et al. (2008) [17]	2+	To observe how sailors manage their sleep and alertness before and during competition in a long-distance regatta	Eight sailors on yachts during the race Tour de France à la Voile (Atlantic and Mediterranean)	2002	Sleep log			PSG: Brainwalker (Medatec software, Braine-le-Château, Belgium) Night Cap: REM view (Respironics, Inc., Bend, OR, USA)
Lützhöft et al. (2010) [6]	2+	To investigate the degree of fatigue on board and compare 6:6 with 4:8 shift schedules	30 watchkeeping nautical officers on 13 Swedish cargo ships (bulk carriers, car carriers, and tankers; 2300 to 11,000 DWT)	2005–2007	KSS	Actiwatch (Cambridge Neurotechnology Ltd., Cambridge, UK)	6-min (RTT-type not specified)	EOG
Matsangas and Shattuck (2018) [10]	2−	To assess similarities and differences between subjective reports of fatigue/sleepiness; to assess predictors of sleepiness/fatigue; to measure sleepiness/fatigue	767 crew members (predominantly watchkeepers) on a U.S. Navy aircraft carrier (NIMITZ, CNV-68)	Spring 2014	ESS FSS			
Matsangas and Shattuck (2020) [2]	2+	To assess the prevalence of disruptive factors in the sleep environment; to assess whether these disruptive factors affect sleep and well-being	1269 sailors (661 watchkeepers and 231 non-watchkeepers) on five ships (one Nimitz-class aircraft carrier, one Ticonderoga-class cruiser, three Arleigh Burke-class Flight IIA destroyers)	2014–2017	ESS PSQI POMS ISI Activity log	Motionlogger Watch (Ambulatory Monitoring, Inc., Ardsley, NY, USA) Spectrum Plus (Philips Respironics, Bend, OR, USA)		
Matsangas and Shattuck (2020) [25]	2+	To assess sleep quality and examine whether Pittsburgh Sleep Quality Index (PSQI) scores are influenced by occupational factors and sleep attributes and whether PSQI can predict impaired PVT performance	872 USN sailors (666 watchkeepers and 206 non-watchkeepers) on seven USN “surface combatants” (one Nimitz-class aircraft carrier, one Ticonderoga-class cruiser, and five Arleigh Burke-class destroyers)	Six periods (December 2012, May 2013, June and November 2014, June 2017, December 2017–January 2018)	PSQI Sleep log Activity log	Motionlogger Watch (Ambulatory Monitoring, Inc., Ardsley, NY, USA) Actiwatch Spectrum (Philips Respironics, Bend, OR, USA)	3-min PVT	
Nordmo et al. (2017) [26]	2−	To examine the association between hardiness and reported insomnia symptoms in a maritime military environment	281 sailors, officers, and enlisted personnel on a Royal Norwegian Navy frigate during a 4-month naval deployment to combat piracy in the Gulf of Aden		BSI			
Oldenburg and Jensen (2019) [27]	2+	To assess the prevalence of drowsiness in seafarers during sea passage with a distinction between day workers and watchkeepers	75 day workers and 123 watchkeepers during 18 voyages on 18 different container ships		ESS SSS	SenseWear armband activity monitor (BodyMedia, Inc., Pittsburgh, PA, USA)		Pupillometry
Shattuck and Matsangas (2016) [28]	2−	To assess mood, sleep patterns, daytime sleepiness, and psychomotor vigilance performance during a 5/10 watch	77 Reactor Division (RX) participants on the aircraft carrier USS Nimitz (CVN-68)	10–27 June 2014	ESS POMS Activity log	Motionlogger Watch (Ambulatory Monitoring, Inc., Ardsley, NY, USA) Actiwatch Spectrum (Philips Respironics, Bend, OR, USA)	3-min PVT	
Shattuck and Matsangas (2017) [29]	2−	To assess the impact of sunlight, long working hours, and caffeinated beverages on average daily sleep duration	91 U.S. Navy crew members (65 men) on the aircraft carrier USS Nimitz	3–14 November 2014	ESS Activity log	Motionlogger Watch (Ambulatory Monitoring, Inc., Ardsley, NY, USA)		
Shattuck and Matsangas (2020) [30]	2+	To compare the well-being and sleep of dayworkers and shift workers	804 sailors (78.4% male) on seven U.S. Navy ships		ESS PSQI POMS ISI Activity log	Motionlogger Watch (Ambulatory Monitoring, Ardsley, NY, USA) Spectrum Plus (Philips Respironics, Bend, OR, USA)		
Sunde et al. (2016) [31]	2+	To assess relationships between noise exposure during sleep and actigraphy-derived sleep parameter	72 participants from different occupational groups on board (engineers, navigators, cooks, etc.) on 21 Royal Norwegian Navy ships	April 2012–June 2013		Actiwatch AW4 (Cambridge Neurotechnology Ltd., Cambridge, UK) Actiwatch 2 (Philips Respironics, Bend, OR, USA)		
Thomas et al. (2019) [32]	2−	To investigate the consequences of fatigue and workload associated with increased operational stress	12 senior staff on board a roll-on roll-off ferry in Australia		CSS/SPFS	Unspecified device (Philips Respironics, Bend, OR, USA)	5-min PVT	
Valdersnes et al. (2017) [33]	2+	To investigate the relationship between worries about possible accidents and sleepiness in seafarers; to investigate PsyCap as a protective factor in this context	397 seafarers from a Norwegian company in the offshore oil and gas industry on 22 ships in the North Sea and Southeast Asia	2012	SOFI			
van Leeuwen et al. (2013) [16]	2−	To investigate sleep, sleepiness, and neuro-behavioural performance in a simulated 4:8 watch system and the effects of disrupting a single free watch simulating a condition of overtime work	30 bridge officers (29 men) measured with a bridge simulator at Chalmers University of Technology, Gothenburg, who slept on the passenger ship “Origo”		KSQ KSS KSD		5-min PVT	PSG: Vitaport 3 recorders (TEMEC, Kerkrade, The Netherlands)
Youn and Lee (2020) [34]	2+	To compare the physical activity intensity and sleep patterns under three conditions: (1) moored versus sailing, (2) on-navigation duty and off-navigation duty, and (3) day versus night navigation duty	51 senior naval students (10 female and 41 male) of the navigation department on training vessels of Mokpo National Maritime University in South Korea on three sea voyages			ActiGraph GT9X Link (ActiGraph, Pensacola, FL, USA)		

RTT, reaction time test. SIGN-criteria [18]: 2+ (well-conducted case-control or cohort studies with a low risk of confounding or bias and a moderate probability that the relationship is causal); 2− (case-control or cohort studies with a high risk of confounding or bias and a significant risk that the relationship is not causal).

**Table 2 ijerph-19-00120-t002:** Overview of questionnaires used on board.

Questionnaires	Number of Studies
Fatigue	9
- Swedish Occupational Fatigue Inventory (SOFI) - Profile of Mood States (POMS) - Fatigue Severity Scale (FSS) - Need for Recovery Scale (NFR) - Samn–Perelli Fatigue Scale (SPFS)/Crew Status Survey CSS	3 3 1 1 1
Sleepiness	9
- Epworth Sleepiness Scale (ESS) - Karolinska Sleepiness Scale (KSS) - Stanford Sleepiness Scale (SSS)	6 2 1
Sleep behaviour	8
- Pittsburgh Sleep Quality Index (PSQI) - Insomnia Severity Index (ISI) - Bergen Insomnia Scale (BIS) - Karolinska Sleep Questionnaire (KSQ)	4 2 1 1

**Table 3 ijerph-19-00120-t003:** Strengths and weaknesses of measuring methods for sleep, fatigue, and sleep behaviour in the maritime-specific study setting.

Method	Strengths	Weaknesses	Weaknesses in the Maritime-Specific Setting
Subjective in general	Suitable for large collectives Cost-effective No investigator on board necessary Suitable for long-term studies	Lack of motivation and compliance	
Questionnaires	Different questionnaires for special questions (e.g., fatigue, sleepiness, sleep quality)	Recall bias Use of different cut-off values Risk of social desirability with time contracts or interviewer presence	Designed for land context (ESS contains question on road behaviour)
Activity and sleep diaries	Easy individual adaptation to study design Supplementation of missing actigraphy data	Often poor comparability due to inconsistent format Sleep duration and quality often overestimated	
Objective in general	Independent of motivation, recall bias, social desirability	Study population limited by number of devices Investigator on board necessary	
Reaction Time Test (RTT)/Psychomotor Vigilance Test (PVT)	Cost-effective Gold standard for detecting sleepiness-related vigilance reduction	Only captures effect of sleepiness (Ship) movement and noise can falsify measurements	
Pupillometry	Fast, uncomplicated handling	Only measures effect of sleepiness (Ship) motion, noise, ambient light can falsify measurements	
Electrooculography (EOG)		Motion artifacts Interindividual different blinking characteristics Time-consuming evaluation	
Actigraphy	Cost-effective Suitable for long-term studies Hardly any first-night effect Good agreement with PSG in TST, SOL, SE%	No display of sleep architecture Overestimation of sleep periods Underestimation of the wake stage	Motion artifacts due to strong sea state/high speed
Ambulatory Polysomnography (PSG)	Comparable with stationary PSG (gold standard of sleep diagnostics) Display of sleep architecture Possibly hardly/less first night effect (in contrast to stationary PSG)	Cost intensive Time-consuming measurement and evaluation	

ESS, Epworth Sleepiness Scale; TST, total sleep time; SOL, sleep-onset latency; SE%, sleep efficiency. Sources can be found in the discussion.

## Data Availability

Not applicable.

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
