# Peer review of "Measurement Methods of Fatigue, Sleepiness, and Sleep Behaviour Aboard Ships: A Systematic Review"

_ijerph, 2021, doi:10.3390/ijerph19010120_

Round 1

Reviewer 1 Report

This is a very well-oriented manuscript attempting to provide an overview of the objective and subjective measurement methods of fatigue, sleepiness, and sleep behavior used on board. The authors performed a rigorous search of the literature, analyzed and presented many items of the articles, providingalso detailed comments.  

Regarding the effect of sleepiness, reaction time measurements (RTT, PVT), as well as pupillometry, were suggested by the authors as useful diagnostic tools. For the determination of sleep behavior over longer periods, the use of actigraphs in combination with diaries, were also suggested. However, the authors omitted to make any recommendation regarding the existence or absence of any subjective or objective methods to validate fatigue. This need to be done.

Although, as the authors themselves concluded, only few measurement-related limitations on board were reported and no disruptions of daily board routines existed, there was a high rate of unreported cases, especially with objective measurement problems. Thus, as the authors admit in the Conclusions Section “it cannot be extrapolated with certainty what impact the maritime environment actually has on the reliability of the measurement methods”.

In my opinion, this is the main problem of this manuscript. In a well-organized and presented manuscript of all published studies, we found only speculations of methods that could be suggested as appropriate. However, this is the only article, to date, that has collected, analyzed and presented all these disparate data.

Author Response

“Regarding the effect of sleepiness, reaction time measurements (RTT, PVT), as well as pupillometry, were suggested by the authors as useful diagnostic tools. For the determination of sleep behavior over longer periods, the use of actigraphs in combination with diaries, were also suggested. However, the authors omitted to make any recommendation regarding the existence or absence of any subjective or objective methods to validate fatigue. This need to be done.”: we agree that the fatigue measurements have been subordinated in our conclusions. Therefore, we have reiterated in the revised manuscript that fatigue can only be measured subjectively and have provided the two most used questionnaires for fatigue as well as for sleepiness in our review studies for orientation (Lines 465-469).

“Although, as the authors themselves concluded, only few measurement-related limitations on board were reported and no disruptions of daily board routines existed, there was a high rate of unreported cases, especially with objective measurement problems. Thus, as the authors admit in the Conclusions Section “it cannot be extrapolated with certainty what impact the maritime environment actually has on the reliability of the measurement methods”. In my opinion, this is the main problem of this manuscript. In a well-organized and presented manuscript of all published studies, we found only speculations of methods that could be suggested as appropriate. However, this is the only article, to date, that has collected, analyzed and presented all these disparate data.”: we agree that this fact is indeed a major limitation of our study. For this reason, we have highlighted this point in the limitations of the reviewed manuscript (Lines 454-458).

Reviewer 2 Report

The article is very precisely processed and focused on a systematic review dealing with measurements of fatigue, sleepiness, and sleep behavior abroad ships. The authors went through 166 studies in which only 21 were selected and complied with authors´ requirements to be enrolled in the study. A combination of subjective and objective methods was used more frequently, and the best combination revealed actigraphy accompanied by reaction time test and diary. The review shows a deep knowledge of authors who devoted a lot of years to this interesting topic and present complex data collected for 11 years from the whole literature (66 references). The review is completed with two detailed Tables summarizing different methods used for evaluation of sleep, fatigue, and sleep behavior in the maritime and their advantages and disadvantages.,

I have only minor remarks:

Introduction: Fatigue and sleepiness are surely two different features. Multiple sleep latency test (MSLT) used in most sleep labs clearly differentiates these two symptoms.

Discussion: Actigraphy should by optimally combined every time with diaries, and 7 days and more is usually recommended as the best actigraphy monitoring time (ICSD-3, 2014).

Author Response

“Introduction: Fatigue and sleepiness are surely two different features. Multiple sleep latency test (MSLT) used in most sleep labs clearly differentiates these two symptoms.”: we agree that only sleepiness can be objectively measured via MSLT and therefore a distinction of these symptoms is possible. To avoid any confusion, we have mentioned this aspect in the introduction (Lines 69-71).

“Discussion: Actigraphy should by optimally combined every time with diaries, and 7 days and more is usually recommended as the best actigraphy monitoring time (ICSD-3, 2014).”: we agree and have included this statement in our revised manuscript in the discussion on optimal actigraphy duration (Lines 345-347).
